# Consistency between Definition and Reasons for Applying Corporate Social Responsibility: The Perspective of Social Responsibility Managers

Oscar Licandro [1], Luis Camilo Ortigueira Sánchez [2] and Oscar Huapaya-Huertas [3,*]

1   Cathedra of Organizational Social Responsibility, Centro Latinoamericano de Economía Humana, Universidad CLAEH, Montevideo 11100, Uruguay; olicandro@claeh.edu.uy
2   Department of Administration, Faculty of Business, Universidad del Pacífico, Jesús María, Lima 15072, Peru; lc.ortigueiras@up.edu.pe
3   Facultad de Ciencias de la Salud, Escuela de Medicina Humana, Universidad Científica del Sur, Lima 15067, Peru
*   Correspondence: ohuapaya@cientifca.edu.pe; Tel.: +51-981063835

**Abstract:** Knowledge about the motivations of managers to practice corporate social responsibility (CSR) is a critical issue for those who promote its adoption. The understanding of these reasons is complicated by the fact that there are different ways of defining CSR, raising the question of whether there is any relationship between the reasons for adopting it and how it is defined. To address this issue, this research categorizes these reasons and relates them to a classification of the different ways of defining CSR. To this end, a self-administered questionnaire was applied to a non-probability sample of social responsibility managers, which included indicators for both classifications. It was found that these managers present all types of motives identified, that proactive motives outweigh reactive motives, and that there is a significant degree of correlation between the ways of defining CSR and the reasons for doing so. From these results, it can be concluded that managers present consistency between their objectives (motives) and means (social responsibility practices).

**Keywords:** corporate social responsibility; motives; definition; social responsibility managers; Uruguay

## 1. Introduction

Since first entering the corporate and academic agenda in the 1950s, the issue of social responsibility among entrepreneurs and companies has gained in importance, giving rise to an extensive academic and institutional literature. Indeed, academic production on corporate social responsibility (CSR) has been prolific, and it has not lost any of its relevance over the last seven decades; so much so that CSR "has established itself as a very active field of study in the discipline of management" [1] (p. 919).

While much research has been conducted on the benefits (results) of CSR, the issue of why business leaders embark on CSR has been little studied. It appears to be assumed that the motivation for CSR is limited solely to the achievement of these benefits and therefore responds only to a type of instrumental rationality. As such, other types of motivation are discarded a priori, such as the moral and religious values of managers, their personal convictions about how their companies should relate to stakeholders, or, more generally, their ideas about the company's goals (in particular, those relating to the role that companies must play in society and in tackling major social problems, such as poverty, human rights, or sustainability). On the other hand, given that the accumulated knowledge about the benefits of CSR is important for decision-making at the micro level of each company, it is not enough to understand CSR as a macro-phenomenon in today's society and in the market economy. It may be that this situation can be explained by the predominant idea that "corporate social responsibility is not located in the realm of social

ends, but of organizational media" [2] (p. 107). A better understanding of motivations for CSR is a useful input for stakeholders interested in promoting the practice—particularly governments, social organizations, universities, and international organizations. Therefore, it is necessary for academia to study these motivations.

In this context, the first objective of the present study is to map the reasons or foundations for applying CSR present in the literature, classify them into homogeneous categories, and evaluate their relative weight in the motivations of managers responsible for CSR in their companies. This mapping is based on some of the few publications that have studied this topic from a theoretical point of view, particularly those that have proposed a taxonomy of the reasons or foundations for applying CSR. This task is complicated by the lack of theoretical consensus on the definition of CSR, which translates into the different interpretations of the concept present in the corporate world. Can the same motives lead a manager to incorporate CSR when they interpret it exclusively as their company's collaboration in solving its own problems (or reduce it to a form of marketing), define CSR as a form of management based on ethical precepts, or interpret it as a way of managing all the company's externalities responsibly?

The second and main objective of this study is to investigate the relationship between the way managers interpret CSR and the motives that lead them to adopt the practice. The intention is to answer the question: is there a relationship between the way managers interpret CSR and the reasons they incorporate it into their companies? This question is important because in management there must be consistency between the actions carried out and the objectives they are intended to achieve. In the application of CSR, actions are determined based on the way managers interpret this concept and the objectives are chosen based on the reasons.

To this end, a self-administered questionnaire was applied to a non-probability sample of CSR managers in Uruguay. Two groups of indicators, evaluated using Likert scales, were used to measure three types of reasons for incorporating CSR and five ways of defining it. Statistical correlation tests were then applied to study the relationship between the two groups of indicators. Finally, the ways of interpreting CSR and the types of motives for adopting the practice were constructed based on a review of studies on CSR, most of which can be considered classics of this literature.

## 2. Literature Review

### 2.1. CSR Managers and CSR Implementation

For several decades, academics have been investigating the role played by managers in the implementation of CSR. A few years after this concept emerged on the corporate agenda, Hay and Gray (1974) [3] observed that managers were evolving from an exclusive orientation to maximizing profitability to an understanding of responsibility not only to owners but to taxpayers and society as well. Wood (2010) [4] noted that the United States Committee for Economic Development (CED) urged "business leaders to contribute to social wellbeing in ways beyond the provision of jobs, taxes, and goods and services" [4] (p. 52). According to Grit (2004) [5], in this new context "managers have to reconcile two mostly contradictory repertoires" [5] (p. 98): on the one hand, generating profits; and on the other, helping to solve societal issues and deciding on the importance of different groups of stakeholders. Amin (2016) [6] argued that "the sense of Ethical Leadership (EL) should take place and can play a critical role to improve the actual impact of the CSR" [6] (p. 77). Conversely, when leaders act by prioritizing their own interests (agency problem) or when they are oriented towards concentrating power, they will likely have less interest in incorporating CSR [7].

Weaver, Treviño, and Cochran (1999) [8] proposed that managers are the actors who can have an effective vision of CSR and the ability to promote it in their companies. In the particular case of middle management, Godkin (2015) [9] highlighted that beliefs, values, and attitudes aid or impede the implementation of CSR, since this application requires ethical leadership that engages employees. The institutionalization of CSR within each

company requires the commitment of senior management because it is a process of cultural change [10]. Guarnieri and Kao (2008) [11] pointed out that managers have the initiative to implement CSR. Mamic (2005) [12] indicated that they are the internal agents that ensure the successful implementation of CSR because, in accordance with Bohem (2002) [13], managers "create opportunities to move in new directions and [...] facilitate change, and thus they use the organization's resources to achieve their aims" [13] (p. 172), as required by CSR. In addition, due to the hierarchical place they occupy in the organization, managers (and particularly middle managers, as is the case of CSR managers) have the possibility to create an environment that favors the implementation of socially responsible behaviors [14].

Miles, Munilla, and Darroch (2006) [15] argued that the successful implementation of CSR depends on having a proper dialogue with stakeholders, and they highlighted the role of top management and middle management in that dialogue. Along similar lines, the ISO 26.000 Guide [16] analyzes the role of business leaders in the internal promotion of CSR, as well as the incorporation of governance mechanisms that ensure their internalization in decision-making processes and in effective communication related to CSR, both within and outside the organization. However, McWilliam and Siegel (2000) [17] warned that some managers have resisted the incorporation of CSR, "arguing that additional investment in CSR is inconsistent with their efforts to maximize profits" [17] (p. 603). Given the necessity of managers' commitment for the successful incorporation of CSR, academics have striven to demonstrate that there is a link between the social and economic performance of companies. Such efforts "were undertaken largely in the hopes of establishing a positive relationship that might be persuasive to business leaders who were skeptical" [4] (p. 60) on CSR.

Galbreath (2016) [18] proposed that companies need internal actors to guide them on the appropriate mechanisms to improve their CSR, and that one way to do this is by appointing senior CSR managers. He argued that these managers help to raise awareness about the aspects of CSR that involve all business processes, encourage positive attitudes towards CSR at lower levels, and tend to focus on social and environmental issues. Argandoña, Fontodrona, Pin, and Lombardia (2008) [19] indicated that the effective incorporation of CSR in companies requires an understanding that it is transversal to the company's operation. Accordingly, they proposed that "the person responsible for CSR, whatever the name given in each company [...] is undoubtedly faced with the difficult task of introducing this criterion into the daily management of the company" [19] (p. 3). For [20], the incorporation of CSR creates the need for an internal leadership, which for different reasons does not usually fall on the classic functional managers, and this is the reason why many companies have created specific CSR managers.

Argandoña et al. (2008) [19] found that in Spain, this function is given different names (such as "CSR director", "CSR manager", and "CSR coordinator"), which in turn indicates that it can be located at different hierarchical levels. Duque, Ortíz, and Arciniegas (2014) [21] showed that the position of CSR manager is emerging in Colombia, and that it is a position more frequently assigned to women than to men. Galbreath (2016) [18] demonstrated that the existence of a CSR manager correlates positively with the application of CSR. In his opinion, these managers enhance the presence of other factors also associated with CSR, such as the integration of women on the boards and the existence of external directors. [20] found that companies with a CSR manager perform better in the application of CSR, but that the difference was not statistically significant, most likely because of the small sample used in their research.

It has also been noted that among managers, there is ignorance and/or incomplete understanding about what social responsibility entails [22], and that "they do not have the necessary experience (social skills) to make socially oriented decisions" [23] (p. 88) because they are more oriented towards finance or operations. Therefore, since the application of CSR is a decision made by managers and owners, it is essential to identify their perceptions about the concept [24]: that is, how they define it and their reasons for adopting it. Although there is published research on the perception of CEOs and managers of specific departments

(human resources or marketing, for example) about CSR, few papers have analyzed the perceptions of dedicated CSR managers, and fewer still explore their motivations for applying it.

### 2.2. The Reasons for the Application of CSR

Several decades ago, academics first pointed out the problems associated with the multiplicity of motivations for incorporating CSR, as well as the lack of research on this problem: "The whole question of motivation with respect to social responsibility is a dense thicket of conjecture" [25] (p. 7). Thus, with the aim of constructing a classification of motivations for CSR, this section analyzes the content of six academic publications that address the topic. Two of these publications are based on a literature review [26,27], and the remaining four result from the authors' own reflections [28–31].

In a widely cited article, Garriga and Melé (2004) [26] analyzed the *theories* underlying the arguments for CSR proposed by a select set of authors, including some classics in this field of study. They concluded that four groups of theories exist, which they labeled instrumental, integrative, ethical, and political. Previously, Moir (2001) [28] proposed that the theories used to analyze and explain CSR were of four types: social contracts, stakeholders, legitimation, and social performance. Porter and Kramer (2006) [29] identified four arguments in favor of CSR, which they termed predominant justifications: moral obligation, sustainability, license to operate, and reputation. Argandoña (2011) [30] proposed that there are five types of arguments used to legitimize CSR: legal case, social case, moral case, business case, and management case. Aguinis and Glavas (2012) [27] addressed the reasons for applying CSR using the term "predictors" to explore the background to CSR actions and policies. According to Aguinis and Glavas, there are at least eight predictors: regulation, standards, sense of responsibility, acting according to a higher order, managers' personal commitment, stakeholder pressures, certification, and instruments (achieving business objectives such as competitiveness or legitimacy). Finally, it is worth mentioning Dare (2016) [31], who classified the reasons as instrumental (to produce results favorable to the company), relational (to build, maintain, or restore the legitimacy of the company in the view of its stakeholders), and moral (to apply CSR driven by the higher purpose of helping humanity).

In the framework of this research, it was decided to group the business managers' motives for incorporating CSR into three broad categories, similarly to [31]. The first includes reasons that have a *normative* **character**; that is, those that are related to the duty to be morally and socially correct. This encompasses the social contract of [28], the ethical and political theories of [26], the moral justification and sustainability of [29], the moral case and the legal case of [30], the first five predictors of [27], and the moral motives of [31]. The second category groups together *integrative* reasons; that is, CSR as the consideration of the needs, expectations, and demands of society, whether certain stakeholders or all of them. This group includes the stakeholder theory of [28], the integrative theories of [26], the social case of [30], the pressures of the stakeholders of [27], and the relational motives of [31]. Finally, the third category contains the various *instrumental* reasons; that is, those directly related to the achievement of a company's objectives. Here, the theories of legitimization and social performance of [28], the instrumental theories of [26], the license to operate and the construction of reputation of [29], the certification and the instrumental predictor of [27], and the instrumental motives of [31] come into play. Notably, Amato, Buraschi, and Peretti (2016) [32] highlighted the instrumental character of CSR, stating that this concept was born as a management tool at the service of sustainability. Examples of motives for each of these three categories are provided in the fifth column of Table 1.

Given that the type of reason for applying CSR is likely to influence CSR strategies and actions, it is necessary to study the motivations of the managers that determine them. Therefore, in this paper the following hypotheses are proposed:

**Hypothesis 1:** *The three types of reasons for the adoption of CSR identified in the literature are present in the motives of CSR managers to apply them.*

**Hypothesis 2:** *The three types of reasons for the adoption of CSR are independent of one another. This means that in general, each CSR manager tends to subscribe to only one of these reasons.*

*2.3. Ways of Defining CSR*

Although some authors suggest that the origins of CSR date back to the 19th century [33], there is a broad consensus that the concept emerged after the Second World War. Despite the time that has elapsed since then, there is still no commonly recognized or agreed definition of CSR, despite the efforts that have been made to homogenize the concept. Many studies have testified to the plurality of ways of interpreting CSR. Eibirt and Parket (1973) [25] warned that it is not easy to establish the contents of CSR or set its limits, while Montiel (2008) [34] pointed out that the plurality of definitions is a reflection of the ambiguous nature of the concept. In the context of Latin America, Carrillo Montoya, Urrea Zazueta, Tereso Ramírez, and Verdugo (2022) [35] noted that the definition of CSR "is under constant construction, both historically as well as theoretically and methodologically" [35] (p. 354). The problem becomes even more complex when it is argued that social responsibility is a model applicable to all types of organizations [36], including public or state organizations [37]. In sum, as Seehi (2015 [38] stated, "the definition of CSR is both complex and complicated" [38] (p. 625).

Despite this conceptual diversity, few academic studies analyze and/or systematize this diversity of definitions. From the literature review, it is possible to identify three groups of papers that address CSR: (1) those that identify the thematic areas or dimensions of CSR (e.g.,: [39–45]); (2) those that study the content of the main definitions proposed in the literature from a historical perspective (for example: [28,34,46–49]); and (3) those that propose some kind of classification of the definitions [50,51]. Taking the academic publications in the third group as a reference, the present study proposed five ways to define CSR: (1) corporate citizenship (or contribution of the company to society); (2) orientation towards stakeholders; (3) ethical management; (4) responsible management of externalities; and (5) corporate social marketing. Examples of each are presented in the third and fourth columns of Table 1 below.

This conceptual plurality justifies the need to ascertain how CSR managers interpret CSR, since CSR-driven strategies and policies must surely be influenced by this interpretation. Managers who interpret CSR as corporate citizenship are likely to emphasize actions both inside and outside the operation of the business itself (such as donations and other philanthropic activities), while those who interpret it as guidance to the stakeholders will focus more on identifying and meeting their demands and needs. It is useful, then, to determine whether the five types of definitions are adhered to by CSR managers in practice, and whether each manager adheres to a single way of defining the concept or takes into account more than one. On the basis of these considerations, the following hypotheses will be tested:

**Hypothesis 3:** *The five ways of interpreting CSR identified in the literature are present in how CSR managers interpret this concept.*

**Hypothesis 4:** *The five groups of CSR definitions are independent of one another. This means that, in general, each CSR manager interprets the concept in alignment with just one of these definitions.*

Few studies have focused on exploring how individual managers define or interpret the concept. Of those that have, Johnston and Beatson (2005) [52] applied the open-ended question technique to a sample of managers, asking "how do you define or what do you understand by the term 'corporate social responsibility'" [52] (p. 3). They found that the responses varied greatly among the respondents, that it was difficult to formulate a structured definition, and that the common denominator was to highlight the voluntary

nature of CSR, which is not a definition in the strict sense. Other studies have used similar techniques. Licandro et al. (2019) [51] applied theirs to a sample of SME managers, while Ferreira et al. (2020) [44] applied it to postgraduate management students from three countries.

There is a long list of studies on the "perceptions" of managers about CSR. But this word is not used to study definitions or interpretations of the concept. For instance, managers have been asked about their main social responsibilities [53]; their opinion on items related to the implementation of CSR [54]; the objectives of their companies in relation to CSR [55]; to evaluate the CSR of their companies [56]; or about how their companies apply CSR practices [57,58]. However, few studies on "perceptions" focus on understanding how managers define or interpret the concept of CSR. And for the specific case of CSR managers, no studies have been found.

### 2.4. The Relationship between Reasons for Applying CSR and Ways of Defining the Concept

In order to compare, from a theoretical point of view, the reasons for incorporating CSR with the different ways of defining it, the present study sought to identify previous studies whose authors adopt positions on each of these two issues. It is not easy to find studies of this type, because few articles explain the definition on which the research is based. Moreover, among those that make this definition explicit, only a small number discuss the reasons for applying it. Table 1 includes papers whose authors propose a CSR definition and rationale. They are classified according to the two variables: type of motives and type of definition. The table shows that each way of defining CSR is based on different types of argument, with the exception of CSR understood as corporate social marketing, which is only based on instrumental arguments.

If academics associate different types of fundamentals with different ways of defining CSR, it is to be expected that the same situation will be observed among CSR managers. If this is the case, it is reasonable to assume that in the application of CSR there will be a degree of inconsistency between purposes (reasons) and actions chosen to achieve those purposes, as the latter are likely to be determined by how CSR is interpreted. This gives rise to the next hypothesis to be tested:

**Hypothesis 5:** *How CSR managers define CSR is independent of the reasons for incorporating it.*

**Table 1.** Classifications of CSR in the literature according to the definition and reason proposed by the authors.

| CSR as: | Type of Argument | Author | Definition | Reason |
|---|---|---|---|---|
| Corporate citizenship | Normative | Van Marrewijk [59] | "CSR refers to company activities—voluntary by definition—demonstrating the inclusion of social and environmental concerns in business operations" [59] (p. 102) | "The motivation for CS is that CS is perceived as a duty and obligation, or correct behavior" [59] (p. 102) |
| | Integrative | Carroll [60] | "It is suggested here that four kinds of social responsibilities constitute total CSR: economic, legal, ethical, and philanthropic" [60] (p. 40) | "[...] here is a natural fit between the idea of corporate social responsibility and an organization's stakeholders. The concept of an interested party personalises social or corporate responsibilities when outlining the specific groups or persons that the company should consider in its CSR orientation" [60] (p. 43) |
| | Instrumental | Porter and Kramer [29] | "More strategic CSR occurs when a company adds a social dimension to its value proposition, making social impact an integral part of the strategy" [29] (p. 13) | "Companies are called to tackle hundreds of social problems, but only a few are opportunities to make a real difference to society or to confer a competitive advantage" [29] (p. 15) |

**Table 1.** *Cont.*

| CSR as: | Type of Argument | Author | Definition | Reason |
|---|---|---|---|---|
| Ethics-based management | Normative | Davis [61] | "The substance of social responsibility arises from concern for the ethical consequences of one's acts as they might affect the interests of others." [61] (p. 46) | "The idea of social responsibility, however, requires him to consider his acts in terms of a whole social system and holds him responsible for the effects of his acts anywhere in that system" [61] (p. 46) |
| | Integrative | Valentine and Fleischman [62] | "A natural extension of organizational ethics is a company's involvement in CSR, which involves answering the requirements of stakeholders, with particular focus on societal issues and challenges" [62] (p. 161) | "[...] social responsibility creates a symbiotic relationship based on "give and take" between stakeholders and companies" [62] (p. 159) |
| | Instrumental | Wadhwa and Bhargava [63] | "[...] corporate social responsibility is the company's constant commitment to ethical behavior" [63] (p. 863) | "Corporate Social Responsibility provides the company with important benefits in risk management, cost savings, capital acquisition, customer relations, and human resource management and innovation capabilities" [63] (p. 864) |
| Guidance to stakeholders | Normative | Johnson [64] | "Instead of striving only for larger profits for its stockholders, a responsible enterprise also takes into account employees, suppliers, dealers, local communities, and the nation" [64] (p. 50) | "[...] business takes place within a socio-cultural system that outlines through norms and business roles particular ways of responding to particular situations and sets out in some detail the prescribed ways of conducting business affairs" [64] (p. 51) |
| | Integrative | Carroll [60] | "It is suggested here that four kinds of social responsibilities constitute total CSR: economic, legal, ethical, and philanthropic." [60] (p. 40) "At its most fundamental level, this is the obligation to do what is right, just, and fair, and to avoid or minimize harm to stakeholders employees, consumers, the environment, and others)" [60] (p. 42) | "The concept of stakeholder personalizes social or societal responsibilities by delineating the specific groups or persons business should consider in its CSR orientation" [60] (p. 43) |
| | Instrumental | Burke and Logdson [65] | "[...] thus serving more fully the interests of stakeholders and society" [65] (p. 495) | "[...] produces substantial business-related benefits for the enterprise, in particular by supporting the core business activities and thus contributing to the effectiveness of the enterprise in fulfilling its mission" ([65] (p. 496) |
| | | García, Azuero, and Peláez [66] | "It is conceived as a business management approach aligned with the objectives and development plans of the organizations, and with the needs and expectations of its stakeholders" [66] (p. 85) | "CSR, driven by a strategic business vision [...] contributing to improving the competitive and sustainable situation of the company" [66] (p. 87) |

**Table 1.** *Cont.*

| CSR as: | Type of Argument | Author | Definition | Reason |
|---|---|---|---|---|
| Externality management | Normative | Davis and Blomstrom [67] | "Businessmen apply social responsibility when they consider the needs and interest of others who may be affected by business actions" [67] (p. 12) | "Social responsibility, therefore, refers to a person's obligation to consider the effects of his decisions and actions on the whole social system" [67] (p. 12) |
| | Integrative | ISO [16] | "[…] the responsibility of an organization for the impacts that its decisions and activities (products, services and processes) cause on society and the environment […]" [16] (p. 106) | "An organization should respect, consider and respond to the interests of its stakeholders" [16] (p. 119) |
| | Instrumental | Fitch [68] | "[…] the serious attempt to solve social problems caused wholly or in part by the corporation" [68] (p. 38) | "[…] to identify and solve those social problems in which they are intimately involved, and when the possibility of profit is available as an incentive" [68] (p. 45) |
| Corporate social marketing | Instrumental | Lichtenstein, Drumwright, and Braig [69] | "[…] the various forms of company involvement with charitable causes and the nonprofits that represent them" [69] (p. 16) "[…] as collaborative marketing relationships flourish in various forms (e.g., CSR initiatives, cobranding, cross-promotions, strategic alliances)" [69] (p. 17) | "[…] is based on the premise that when a company visibly aligns with a nonprofit, consumers may reasonably infer that support of the nonprofit is also support of a goal of the corporation with which they identify" [69] (p. 17) |

Source: own elaboration.

## 3. Methodology

Universe and sample. The universe is made up of managers who lead CSR activities in companies operating in Uruguay. These companies are a small and unquantified proportion of the total number of companies with a presence in the country. Some of these managers lead the entire CSR process, but others are focused solely on managing CSR activities aimed at the community. The sample is non-probabilistic, obtained from a database built from different sources of information: the websites of companies in the universe, references from social organizations that work collaboratively with companies and from associations promoting CSR, and press articles. A questionnaire was administered to the 178 companies in the database, 56 of which responded, giving a response rate of 31.4 per cent. It is important to note that the questionnaire was devised specifically for this research and that it was not previously validated. Each respondent chose whether to complete the questionnaire via self-administration (sent by email) or personal interview. Once the quality of the questionnaires was evaluated, five were eliminated (three questionnaires were incomplete and two questionnaires corresponded to two companies that do not practice CSR).

Indicators and measures. The motives were operationalized using six indicators, corresponding to two indicators for each of the three types of motives: ethical, integrative, and instrumental (see Table 2). Each of the indicators consisted of a phrase beginning with the causal conjunction "Because". As the variable reasons for applying CSR have "rarely been studied using quantitative methods" [31] (p. 100), the phrases were chosen arbitrarily. To measure the responses, a five-value Likert scale was used, in which 1 corresponds to "do not agree at all" and 5 corresponds to "totally agree." The definitions of CSR were operationalized using the five indicators (using one indicator for each type of definition). These indicators were written in such a way that they express each of the ways of defining CSR found in the literature, and they were measured using the same Likert scale employed to measure the motives (see Table 2).

## 4. Results

Each reason was operationalized by way of a proactive and a reactive indicator. Thus, for example, "incorporating CSR because it is a moral obligation" indicates a reactive attitude to the perception of an external requirement, whereas "incorporating CSR because it reflects the managers' personal convictions" denotes a proactive attitude born of self-initiative. Table 3 presents some descriptive statistics relating to the indicators on the reasons for applying CSR: mean, standard deviation, the percentage that responded "totally agree", and the percentage that answered "do not agree at all." It can be seen that the proactive motive received greater adhesion than the reactive motive for all three groups of reasons. This suggests that the incorporation of CSR stems mostly from the interest and initiative of the managers, rather than the perception of external factors to respond or adapt to (moral obligations, stakeholder demands, or prevailing trends).

On average, the two indicators relating to ethical motivation were those for which the highest scores were recorded, followed by those on integrative motivation. Meanwhile, with much lower scores, the two indicators on instrumental motivations are placed as average. With the exception of the INS2 indicator ("because it is fashionable"), which amounts to a highly negative argument for a manager, the indicators corresponding to the other reasons were endorsed by an average or high proportion of the sample. For this reason, Hypothesis 1 is validated: the three groups of reasons behind the adoption of CSR identified in the literature are present in the fundamentals of CSR managers.

**Table 2.** Descriptive statistics of each type of foundation.

| Type of Justification (Reason) | Cod | Indicator | Responded | | | |
| --- | --- | --- | --- | --- | --- | --- |
| | | | Does Not Match at All | Fully Agreed | Average | Standard Deviation |
| Ethical | ETH1 | Because it is a moral obligation of every company. | 11.8% | 31.4% | 3.49 | 1.377 |
| | ETH2 | Because it is part of the personal convictions of those who lead the company. | 2.0% | 54.9% | 4.27 | 1.002 |
| Integrative | INT1 | Because it helps to better manage stakeholder relations. | 3.9% | 37.3% | 3.86 | 1.096 |
| | INT2 | Because it becomes a requirement of some important stakeholders for YOUR company (for example: customers, employees, the state, or others). | 17.6% | 13.7% | 3.02 | 1.304 |
| Instrumental | INS1 | Because it contributes to a good image of the company. | 11.8% | 7.8% | 3.06 | 1.156 |
| | INS2 | Because it is fashionable and we must not be left behind. | 64.7% | 3.9% | 1.67 | 1.071 |

Table 3 presents the correlation coefficients that result from applying an ANOVA test to the six indicators corresponding to reasons for incorporating CSR. Some interesting results can be observed. First, the pair of indicators on the type of ethical motivation strongly correlate with each other and the pair of indicators on instrumental motivation also correlate (although less strongly), while the pair of indicators on integrative motives have a low correlation (and with a lower significance level: 90%). Second, ethical reasons are independent of other reasons. Third, there are correlations between indicators corresponding to integrative and instrumental motives, but in opposite directions: they tend to be positive when the instrumental argument is the contribution of CSR to good company image, but they tend to be negative when instrumental argumentation is related to not being left out of prevailing trends in the corporate world. The incorporation of CSR as a reactive behavior in response to stakeholder demands is aligned with its incorporation as a brand management tool (proactive behavior), but not with incorporation to adhere to trends

(reactive behavior). The proactive integrative argument (CSR helps to better manage the relationship with stakeholders) does not correlate with the proactive instrumental motive (CSR serves to improve the image of the company), while negatively correlating with the reactive instrumental argument (because it is trendy).

In sum, the results show that: (1) ethical motives are independent of the other two types of motives, and (2) there are some interdependencies between integrative and instrumental motives. Therefore, these results partially invalidate Hypothesis 2 as formulated: the three types of reasons for the adoption of CSR are independent of one other. This means that in general, only one of these reasons tends to predominate for each CSR manager, but there is a degree of coexistence between integrative and instrumental reasons.

**Table 3.** Correlation between types of CSR rationale.

| Type of Justification (Reason) | Cod | Type of Justification (Reason) | | | | | |
| | | Ethic | | Integrative | | Instrumental | |
| | | ETH1 | ETH2 | INT1 | INT2 | INS1 | INS2 |
| Ethical | ETH1 | 1 | 0.510 ** | −0.008 | −0.050 | −0.056 | −0.009 |
| | ETH2 | | 1 | −0.093 | −0.065 | 0.038 | −0.006 |
| Integrative | INT1 | | | 1 | 0.254 * | 0.228 | −0.278 * |
| | INT2 | | | | 1 | 0.371 ** | 0.205 |
| Instrument | INS1 | | | | | 1 | 0.307 * |
| | INS2 | | | | | | 1 |

**. The correlation is significant at level 0.01 (two tails). * The correlation is significant at a level of 0.10 (two tails).

To evaluate Hypothesis 3, the statistics included in Table 4 were calculated: mean, standard deviation, percentage that answered "totally agree", and percentage that answered "do not agree at all". It can be seen that the ethical definition is present in the vast majority of CSR managers and that, conversely, the definition of CSR as social marketing is supported by a very small proportion of the sample. The interpretation of CSR as management based on considering the needs of stakeholders and building win-win relationships with them is in second place, while the definition aligned with ISO 26000 is subscribed to by barely half of the managers surveyed. These results validate Hypothesis 3 (the five ways of interpreting CSR identified in the literature are present in the way CSR managers interpret this concept) but reveal the predomination of definitions associating CSR with ethical management, responding to stakeholders, and giving back to society what the company receives from it.

The results in Table 4 suggest that some managers agree with more than one of the definitions. However, as can be seen in Table 5, the definitions adopted by them are almost all independent of each other. Only (1) an important correlation between the definitions of CSR as attention to stakeholders and as management of externalities and (2) a weak correlation between the definitions of CSR as ethical management and as corporate citizenship (contribution to society) are present. These results invite some interpretations. The first suggests that the definitions of CSR as stakeholder care and externality management are complementary: while the former stresses directly addressing the needs of and building relationships with stakeholders, the latter emphasizes management of the operational externalities on them. Moreover, given that the definition of CSR as ethical management is expressed by 90% of respondents, and that this definition does not correlate with any of the other definitions—besides weak correlation with CSR as corporate citizenship—it is possible to infer that managers include ethics as a dimension of CSR rather than as a definition of the concept in the strict sense. These results amount to a partial validation of Hypothesis 4: that the five types of CSR definitions are independent of each other.

**Table 4.** Descriptive statistics for each type of definition.

| Type of Definition | Indicator | Responded | | | |
|---|---|---|---|---|---|
| | | Do Not Agree at All | Totally Agree | Average | Standard Deviation |
| DEF1 | It is a way of managing the company based on ethical principles. | 3.9% | 90% | 4.51 | 0.784 |
| DEF2 | It is a form of management based on looking at the needs of the company's stakeholders and building win-win relationships with them. | 7.8% | 78% | 4.25 | 1.093 |
| DEF3 | It is a way of giving back what the company receives from society. | 15.7% | 59% | 3.71 | 1.101 |
| DEF4 | It is a desire to reduce the negative impacts of the company's operations on stakeholders and society. | 35.3% | 45% | 3.04 | 1.442 |
| DEF5 | It is mainly a tool for strengthening the company's image. | 58.8% | 22% | 2.47 | 1.120 |

**Table 5.** Correlation between types of CSR definitions.

| | DEF2 | DEF3 | DEF4 | DEFf5 |
|---|---|---|---|---|
| DEF1 | 0.032 | 0.270 | 0.123 | 0.154 |
| DEF2 | 1 | −0.003 | 0.463 ** | 0.096 |
| DEF3 | | 1 | −0.056 | 0.033 |
| DEF4 | | | 1 | 0.075 |

**. The correlation is significant at level 0.01 (two tails).

To evaluate Hypothesis 5, correlation coefficients were calculated between each way of defining CSR and each type of reason to incorporate it. These coefficients are presented in Table 6. The results show that ethical motivations are only associated with the definition of CSR as ethical management. They also show that instrumental motives correlate with the definition of CSR as corporate social marketing, and that only one motive correlates more weakly with the interpretations of CSR as orientation towards stakeholders. On the other hand, integrative motives correlate with the definition of CSR as orientation towards stakeholders and as management of externalities, which is consistent from the theoretical point of view. Indeed, stakeholder orientation presupposes integrative motivation, while responsible management of externalities requires dialogue with stakeholders (integrative orientation) in order to correctly identify the externalities of the operation on them. Finally, it should be noted that the definition of CSR as corporate citizenship is independent of the three types of reasons.

Taken together, these results invalidate Hypothesis 5: how CSR managers define CSR is independent of the reasons for incorporating it. Indeed, only the definition of CSR as corporate citizenship was independent of the reason for incorporating CSR. Conversely, the definition of CSR as ethical management is solely linked to ethical motives, while the definition of CSR as corporate social marketing is only associated with instrumental motives (in particular, the correlation between the definition of CSR as corporate social marketing and the reason related to the construction of brand image is very strong). The definitions of CSR as orientation towards stakeholders and management of externalities represent an intermediate situation, because although both were independent of ethical reasons and (partially) instrumental reasons, each is interdependent with integrative reasons. In sum, the results of this study indicate that coherence between the definition of CSR and the reasons for applying it predominate among CSR managers.

**Table 6.** Correlation between types of CSR definition and types of rationale.

| Type of Justification (Reason) | Cod | Indicator | Interpretation of CSR | | | | |
| | | | DEF1 | DEF2 | DEF3 | DEF4 | DEF5 |
| | | | Ethics-Based Management | Stakeholder Orientation | Corporate Citizenship | Externalities Management | Corporate Social Marketing |
| Ethics | ETH1 | Because it is a moral obligation of every company. | 0.283 * | −0.058 | 0.110 | −0.080 | −0.153 |
| | ETH2 | Because it is part of the personal convictions of those who lead the company. | 0.353 * | 0.008 | 0.202 | 0.062 | −0.010 |
| Integrative | INT1 | Because it helps to better manage stakeholder relations. | −0.057 | 0.648 ** | −0.001 | 0.345 * | 0.070 |
| | INT2 | Because it becomes a demand of some stakeholders. Important for YOUR company (for example: customers, employees, the state, or others). | 0.244 | 0.319 * | 0.143 | 0.500 ** | 0.213 |
| Instrument | INS1 | Because it contributes to a good image of the company. | 0.231 | 0.289 * | 0.232 | 0.155 | 0.704 ** |
| | INS2 | Because it is trendy and the company must not be left behind. | −0.103 | −0.148 | −0.034 | −0.030 | 0.334 * |

**. The correlation is significant at level 0.01 (two tails). * The correlation is significant at a level of 0.10 (two tails).

## 5. Conclusions

Researchers in the field of CSR have focused on the study of the benefits that the application of the concept yields for companies but have paid little attention to the study of the motives that lead managers to incorporate it. In particular, this problematic has not been investigated with reference to CSR managers. Of course, the pursuit of certain benefits is a motive, but it is solely an instrumental motive, supported by a result-oriented rationale. This study found that management decisions around CSR are also based on ethical motives (such as personal convictions or a sense of moral duty) and reasons stemming from a conviction that the company must incorporate stakeholder perspectives into its management.

Another important result is the existence of a certain degree of statistical correlation between the ways of defining CSR and the reasons for incorporating it. This suggests that although managers interpret CSR differently, there is a degree of consistency between how it is interpreted and the reasons for applying it. For example, ethical motivations were found to be associated more with the definition of CSR as ethical management, while integrative motivation is more closely connected with the definition of CSR as management based on the consideration of stakeholders' needs. Since each way of interpreting CSR involves a focus on different types of practices, it can be inferred that, in general, decision-makers choose CSR practices in accordance with the reasons that prompt them to apply these practices. Thus, for example, in the case of instrumental motivations, it would be confirmed that companies that focus on corporate social marketing actions do so strongly based on this type of motive.

The results of this study raise new problems and research questions. Since it was found here that CSR managers do not base their decision on a single type of motive, there is a need for a better understanding of how different types of motives are combined in the minds of these decision-makers. A better understanding of this phenomenon can contribute to the debate between those who understand CSR as something that all companies should adopt and those who propose that its adoption is something that companies choose to do according to corporate objectives. There is also a need to study in greater depth the difference between reactive and proactive motives. Here it was found that proactive motives are more prevalent than reactive ones, regardless of the type of motive. If this is confirmed in other studies, it could be concluded that the adoption of CSR is more proactive

than reactive behavior. This conclusion constitutes a relevant input for the different public, private, and third sector actors that promote CSR.

This research contributes to empirical research on CSR because it provides information on a topic that is rarely addressed by academics. Moreover, the study makes a contribution to the corporate world by highlighting the importance of consistently articulating CSR actions with companies' motives for applying it. Furthermore, it makes a contribution to universities because it shows the importance of teaching students to understand the theoretical and practical complexities of CSR.

The research has some limitations which need to be made explicit. First, the use of non-random sampling and the small sample size suggest that the statistical results are weak and should therefore be considered as merely preliminary. Second, it should be noted that only two indicators were used to represent each type of reason, which clearly indicates that only partial aspects of each indicator were measured. In this regard, future research should build and validate a set of indicators covering the different aspects included in each type of motive. For example, in the case of instrumental motives, indicators should be designed for reasons such as increasing employee engagement, consolidating corporate culture, obtaining operating permission (social leave) from the community, and strengthening corporate reputation. Moreover, in the case of normative reasons, indicators should be included for reasons such as the personal values of decision-makers, the prevailing values in the country where the company operates, corporate culture, the definition of the mission adopted by companies, and the convictions of decision-makers about the role of the company in society and the problems that affect it.

**Author Contributions:** All authors participated in the designed, data collection, and drafting of the article and accepted its final version. All authors have read and agreed to the published version of the manuscript.

**Funding:** The authors received no specific funding for this work.

**Institutional Review Board Statement:** Not applicable.

**Informed Consent Statement:** Not applicable.

**Data Availability Statement:** Not applicable.

**Acknowledgments:** The authors thank the Universidad Científica del Sur for their support in this research.

**Conflicts of Interest:** The authors declare no conflict of interest.

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
