# Peer review of "Consistency between Definition and Reasons for Applying Corporate Social Responsibility: The Perspective of Social Responsibility Managers"

_sustainability, doi:10.3390/su152014838_

Round 1

Reviewer 1 Report

The article is interesting and novel, it is important to make some methodological clarifications that will allow future research to specify information avoiding duplication of effortssuch asfor example, the reasons for excluding some of the data. It is suggested to make some adjustments to the evaluation PDF.

Author Response

We thank the reviewer for his positive comments on the article. We also welcome your suggestions and comments. These suggestions and observations have helped us to improve the article. The changes were introduced in the document, in the same place where the observations were made.

Reviewer 2 Report

*  Overall, the work is holistic ad touches on a current issue of CSR. The research design and flow is good and the research topic is interesting. However the methodology part should be improved. The research methodology and design should be explained clearer. It is too narrow and can not explain the research implemented properly. How the questions developed from the literature? What is the timing of the research?   These sentence is not clear: "Once the quality of the questionnaires was evaluated" "five were eliminated due to different problems" What are the scientific criteria for the "quality" of a questionnaire?   While writing the Hypothesis, including shortly (or in the paragraph) the reasons will clarify the description: "The three types of reasons that justify the adoption of CSR identified in the literature are present in the motives of CSR managers to apply it."  Using only numbers for the citations makes difficult to read and follow the meaning. In some sentences there are different examples of using references: " [25] warned that it is not easy to establish the contents of CSR or set its limits, while Montiel (2008) pointed out that the plurality of definitions is a reflection of the ambiguous nature of the concept." * The academic, managerial and industrial contributions of this particular study should be explained better. What are the contributions and suggestions? 

* The language should be developed. English editing/proofreading is suggested. Some sentences are quite long and difficult to understand. For example: " Can the same reasons lead a manager to incorporate CSR when they interpret it exclusively as the collaboration of their company in solving the problems of the company (or, even when it reduces it to a form of marketing), that when you define it as a form of management based on ethical precepts or when you interpret it as a way of managing responsibly all the externalities of your company?"

Author Response

Responses to reviewer 2

We thank the reviewer for his positive comments on the article. We also welcome your suggestions and comments. These suggestions and observations have helped us to improve the article. Based on most of these suggestions and observations we have made changes to the text.

In relation to three of his observations we comment on what is below. Observation 1. “How the questions developed from the literature?”

Answer. In the case of the indicators on motives, the questions were selected arbitrarily, because we did not find precedents in the literature. We rely on the textual quote from Dare, J. (2016. p.100). This is explicitly mentioned in the document.

Observation 2: “While writing the Hypothesis, including shortly (or in the paragraph) the reasons will clarify the description: "The three types of reasons that justify the adoption of CSR identified in the literature are present in the motives of CSR managers to apply it"

Answer: We have placed the hypotheses in the theoretical framework section. The theoretical discussion prior to each hypothesis provides the reasons why each hypothesis is formulated.

Observation 3: “Using only numbers for the citations makes difficult to read and follow the meaning. In some sentences there are different examples of using references”

Answer: We apply Sustainability standards for citations and references. These standards are very different from those used by other journals (for example: APA).

Reviewer 3 Report

Comments to the Authors

Thank you for the possibility to review the paper. The article is interesting, the researched problem has scientific potential and may be of interest to potential readers. Although I evaluate the article positively, I suggest a few more changes in order to improve it.

Hypotheses Development:

It is essential to add this section to your article separately, which is the most important part of article.

Discussions:

It is essential to add this section to your article separately, and add the following sub-headings.

1-      Theoretical Implications

2-      Practical Implications

3-      Research Limitations and Future Research Directions

Minor editing of English language required

Author Response

We thank the reviewer for his positive comments on the article. We also welcome your suggestions and comments. These suggestions and observations have helped us to improve the article. The changes were introduced in the new version of the document.

In relation to two of your observations, we share with you our responses.

Observation 1: “Hypotheses Development: It is essential to add this section to your article separately, which is the most important part of article”.

Answer: The hypotheses can be included in a specific section or they can be included in the theoretical framework section. Some authors prefer to include the hypotheses in the theoretical framework section, because this makes it easier to understand the relationship between the hypotheses and the theoretical discussion that supports their formulation. This way of locating the hypotheses seems better to us.

Observation 2: “Discussions: It is essential to add this section to your article separately, and add the following sub-headings.1- Theoretical Implications 2- Practical Implications”

Answer: The reflection on the theoretical implications is included in the third paragraph of the conclusions. We choose to formulate these implications as new research questions and problems. We agree with the reviewer that practical implications need to be mentioned. For this reason, we have added a paragraph in the conclusions.

Reviewer 4 Report

The authors analyze the CSR topic by defining 5 hypotheses that they validate through a series of scientific models. A self-administered questionnaire was applied to a non-probability sample of Social Responsibility managers. The results of this paper raise new issues and research questions regarding how managers interpret CSR.

The investigation is quite extensive, and the authors formulate conclusions that lead to validating the hypotheses.

For certain references, the authors do not correctly apply the citation system. For example [3], [4], [5] … and more. Add the authors' names when you refer to a specific paper in the text.

Correct application for reference 3:  … corporate agenda, … Hay and Gray (1974) observed …. and society [3].

 Also, the citations with the page are marked separately:  [1] (p. 919) not [1 p. 919].

Some comments of the authors are present in the paper: CAMBIAR LAS COMAS POR PUNTOS (4 times)

Author Response

We thank the reviewer for his positive comments on the article.

The reviewer questions the way in which we have cited the bibliography. In this sense, the reviewer has indicated: “For certain references, the authors do not correctly apply the citation system. For example *3+, *4+, *5+ ... and more. Add the authors' names when you refer to a specific paper in the text. Correct application for reference 3: ... corporate agenda, ... Hay and Gray (1974) observed .... and society *3+. Also, the citations with the page are marked separately: *1+ (p. 919) not *1 p. 919+”.

Answer. We have applied the Sustainability citation rule, which replaces (Author, year) by [n], where n is the place that the cited document occupies in the list of references. We have corrected the incorrectly worded quotes. In some cases we had quoted like this: [n] (p.x). We have corrected this by ([n] p.x), as indicated in the rules.

Round 2

Reviewer 4 Report

The authors tried to consider the observations of the review process. The previous observation drew attention to the difficult way of reading the manuscript.

The last observation regarding the text “CAMBIAR LAS COMAS POR PUNTOS” has not been resolved.

Since it does not affect the quality of the work but only affects the reading process, we consider that the paper can be proposed for publication in this form after minor revision.

Author Response

We thank the reviewer for his comments.

We recognize that the reviewer is correct when he points out that the comment “CAMBIAR LAS COMAS POR PUNTOS”, was not removed. It was an omission. We have made the correction in the new version of the article, which we sent along with this response